# The Influence of Hip and Knee Joint Angles on Quadriceps Muscle-Tendon Unit Properties during Maximal Voluntary Isometric Contraction

**DOI:** 10.3390/ijerph20053947

**Published:** 2023-02-23

**Authors:** Alessandra Martins Melo de Sousa, Jonathan Galvão Tenório Cavalcante, Martim Bottaro, Denis César Leite Vieira, Nicolas Babault, Jeam Marcel Geremia, Patrick Corrigan, Karin Grävare Silbernagel, João Luiz Quaglioti Durigan, Rita de Cássia Marqueti

**Affiliations:** 1Laboratory of Muscle and Tendon Plasticity, Graduate Program of Rehabilitation Sciences, University of Brasília, Brasília 72220275, Brazil; 2Laboratory of Muscle and Tendon Plasticity, College of Physical Education, University of Brasília, Brasília 70910900, Brazil; 3College of Physical Education, University of Brasília, Brasília 70910900, Brazil; 4Centre d’Expertise de la Performance, INSERM U1093 CAPS, Sports Science Faculty, University of Burgundy, 21078 Dijon, France; 5Exercise Research Laboratory, School of Physical Education, Physical Therapy, and Dance, Federal University of Rio Grande do Sul, Porto Alegre 90690200, Brazil; 6Department of Physical Therapy and Athletic Training, Saint Louis University, St. Louis, MO 63104, USA; 7Department of Physical Therapy, University of Delaware, Newark, DE 19713, USA; 8Laboratory of Molecular Analysis, Graduate Program of Rehabilitation Sciences, University of Brasília, Brasília 72220275, Brazil

**Keywords:** moment-angle relationship, muscle length, mechanical properties

## Abstract

Determining how the quadriceps femoris musculotendinous unit functions, according to hip and knee joint angles, may help with clinical decisions when prescribing knee extension exercises. We aimed to determine the effect of hip and knee joint angles on structure and neuromuscular functioning of all constituents of the quadriceps femoris and patellar tendon properties. Twenty young males were evaluated in four positions: seated and supine in both 20° and 60° of knee flexion (SIT20, SIT60, SUP20, and SUP60). Peak knee extension torque was determined during maximal voluntary isometric contraction (MVIC). Ultrasound imaging was used at rest and during MVIC to characterize quadriceps femoris muscle and tendon aponeurosis complex stiffness. We found that peak torque and neuromuscular efficiency were higher for SUP60 and SIT60 compared to SUP20 and SIT20 position. We found higher fascicle length and lower pennation angle in positions with the knee flexed at 60°. The tendon aponeurosis complex stiffness, tendon force, stiffness, stress, and Young’s modulus seemed greater in more elongated positions (60°) than in shortened positions (20°). In conclusion, clinicians should consider positioning at 60° of knee flexion rather than 20°, regardless if seated or supine, during rehabilitation to load the musculotendinous unit enough to stimulate a cellular response.

## 1. Introduction

The quadriceps musculature weakness has been associated with the initiation, progression and severity of knee osteoarthritis [1]. Thus, strengthening exercise has been identified as a powerful intervention to treat knee injuries, including following surgery. However, pain and arthrogenic muscle inhibition are significant barriers to the generation of adequate stimulus for improvement in muscle function [2,3]. Among the strategies to fasten recovery, isometric training is often used in rehabilitation programs because it may increase muscle force faster than dynamic exercise, with the benefit of lower joint shear stress [4] but the adaptations according to joint angle (i.e., muscle-tendon unit length) raise questions regarding the most appropriate lower limb position according to training objectives.

The quadriceps femoris musculature is mainly responsible for the knee extension torque. The aponeuroses of its four constituents are joined distally to form the quadriceps tendon, and, lastly, the patellar tendon, as the final force-transmission structure [5,6]. Due to origin and insertion characteristics of the quadriceps femoris musculature, changes in both hip and knee joint angles have implications on force production [7] There is a general consensus on the influence of knee joint angle on knee extensor torque during maximum voluntary isometric contraction (MVICs), where knee extension torque is commonly greater at knee flexion angles closer to 60° [7,8,9]. However, exercise prescription for knee extension requires a full reporting of body mechanics, mainly the lower limbs, but the combined effect hip and knee joint angles on force production is still a matter of debate [10]. When comparing the supine and seated hip positions, voluntary torque may be reduced or not change in supine for several knee joint angles (20°–90°) [11,12].

Although positions closer to 60° knee flexion are beneficial for greater force production [8,10], patellofemoral and tibiofemoral compressive forces increase as the knee flexes [13]. In contrast, in supine with the knee fully extended, there is minimal contact between the femur and the patella [14]. Therefore, investigating lower limb positions that reduce joint stress, without reducing force production may be helpful in cases of painful knee, risk of accelerated knee osteoarthritis, or for bedridden patients. Considering that several physiological parameters are involved in muscle force production and the adaptations to exercise, the neural (electrical activity) and morphological (muscle-tendon structure) adaptations must also be elucidated in the context of lower limb position [15].

In shortened positions, the quadriceps femoris musculature has greater activation than in elongated positions [16] and a higher value for activation could be expected in the seated versus supine position for the superficial quadriceps: vastus medialis (VM), vastus lateralis (VL) and rectus femoris (RF) muscles [11] Compared to shortened, mid-range and elongated positions allow greater fascicle length (Lf) [17] and lower pennation angle (θp) [17]. These alterations directly imply that, under isometric conditions, a muscle with longer fascicles may be expected to develop torque more quickly (i.e., higher shortening velocity); and the decrease of the pennation angle, at the elongated positions, implies a mechanical advantage for force generation [17].

Tendon mechanical properties are also of paramount importance for muscle function and integrity [18]. Thus, muscle work can be affected by reduced tendon function due to connective tissue disorders [19], periods of insufficient tendon loading [18,19], or positioning of the joint [20]. Tendon stiffness is greater at longer muscle length than at shorter muscle length, after isometric training [20] and resistance training increase force production capacity, stiffness, and resistance to stress [21], which is beneficial for tissue remodeling. The tendon aponeurosis complex (TAC) stiffness shows the relationship between elongation of the deep aponeurosis to the distal free tendon in response to muscle force to the bones [22]. It is well known that joint angles in elongated positions remove the looseness of the TAC (increasing its stiffness), and optimize muscle length for greater force production, seeming ideal for speeding up adaptation [23,24]. In addition, the tension of the TAC is enlarged in strained situations allowing less muscle work due to better force transmission [25,26,27].

Therefore, we aimed to investigate the influence of hip (0° or 85°) and knee (60° or 20°) joint angles on the knee extensor MVIC, along with surface electromyography (EMG), neuromuscular activity, muscle architecture, TAC stiffness of the four quadriceps femoris constituents, and patellar tendon properties in healthy male subjects. Specifically, we hypothesized that: (1) peak knee extension torque during MVIC would be greater at 60° of knee flexion compared to 20°, as well as greater in seated compared to supine; (2) greater quadriceps femoris muscle activity at 20° of knee flexion compared to 60°; (3) a higher neuromuscular efficiency on 60° positions; (4) at rest and during MVIC, the Lf would be greater, and the θp would be lower when the knee is at 60°; and (5) the TAC stiffness and the patellar tendon stiffness can be higher at more elongated positions in comparison to shortened positions. This study is important because determining how the neuromuscular and muscle-tendon unit acts according to lower limb position may help researchers and exercise prescribers in the rationale for their choices.

## 2. Material and Methods

### 2.1. Trial Design

This was a study with randomized, single blinded, repeated measures. This is a sub-study to a larger trial that is aimed at gaining a better understanding of muscle-tendon adaptation based on hip and knee joint angles. The full protocol is available on ClinicalTrials.gov (Identifier: NCT03822221). Participants were guided regarding the purposes, benefits, and risks before recruitment, and all afforded written consent. Consent was received (protocol number 94388718.8.0000.8093) from the Research Ethics Committee at the University of Brasília/Faculty of Ceilândia following the Helsinki Declaration of 1975. This study was stated according to the Consolidated Standards of Reporting Trials (CONSORT) Statement for Randomised Trials of Nonpharmacologic Treatments [28]. All the procedures were performed in the Laboratory of Strength of the Faculty of Physical Education at the University of Brasília.

### 2.2. Participants

Twenty male participants (age 24 ± 4.6 years; height 177 ± 6.3 cm, and body mass 77 ± 9.3 kg) were recruited through flyers and oral invite. They also had expected values of quadriceps muscle thickness and subcutaneous tissue thickness on the anterior thigh (RF: 24.8 ± 3.97 mm; VL 22.6 ± 2.80 mm; VM 23.9 ± 4.5; vastus intermedius [VI] 19.6 ± 2.5 mm; subcutaneous tissue: 4.32 ± 0.02 mm), which were obtained in supine with 20° of knee flexion. The inclusion criteria were: healthy male, aged 18 to 30 years, and physically active. The exclusion criteria were: involved in regular lower limb strengthening or sports competitions in the prior six months, any musculoskeletal abnormality (including reduced lower limb range of motion, deformity or amputation in any part of the lower limbs; history of patellar dislocation or trauma to limbs or trunk that may interfere in the results), motor control disorder, or systemic diseases that could affect performance or safety on tests.

### 2.3. Randomization, Allocation Concealment, and Blinding

Testing was performed in four positions (Figure 1). Supine and seated were considered 0° and 85° of hip flexion, respectively. A fully extended knee was considered as 0° of knee flexion. Order of the testing positions was randomized for each participant. Randomization was guaranteed by having participants blindly remove four small square paper sheets from an opaque envelope. Participants were also blinded to study aims and hypotheses to prevent expectancy affecting performance. Nevertheless, researchers and volunteers could not be blinded to the assessment positions.

### 2.4. Experimental Procedures

The protocol consisted of five laboratory visits (Figure 1), including a familiarization visit and four experimental visits. Each visit lasted 2–3 h and was performed seven days after the previous visit. We instructed participants to abstain from alcohol and stimulants (e.g., caffeine, chocolate, and performance supplements) for at least 24 h before visits, avert hard exercise 36 h ahead of the visits, and keep their common diet. We obtained anthropometrics (body mass and height) during the familiarization visit, and participants practiced ramped MVICs in each position. The positions were tried apart in each experimental session and composed of 12 MVICs to complete all ultrasound imaging and electromyography exams (two for EMG, two for patellar tendon properties, and eight for muscle architecture).

### 2.5. Outcomes

We measured the peak knee extension torque during MVIC in each position, along with Root Mean Square (RMS) by EMG of the quadriceps musculature. Moreover, the tendon-aponeurosis complex (TAC) stiffness and the muscle architecture (θ_p_ and *L*_f_) were measured from the four quadriceps femoris musculature, and the morphological (cross-sectional area (CSA) and resting length) and, mechanical (stiffness, force, and elongation), and material properties (Young’s Modulus and stress, strain,) were measured from the patellar tendon.

#### 2.5.1. Torque Evaluation

A computerized dynamometer (System 4; Biodex Medical Systems, Shirley, New York, USA) was applied to collect knee extension torque during MVICs of the dominant limb (i.e., the preferred limb to kick a ball). The mechanical axis of the dynamometer was visibly in line with the flexion-extension axis of the knee and hip angles, which were adjusted with a goniometer. The lever arm of the dynamometer transducer was attached 2–3 cm superior the lateral malleolus with a girdle. Subjects were stabilized in the chair using belts on the chest and pelvic girdle to reduce body movement. Seat height was adjusted to each subject’s height to ensure a comfortable fit. Contact of the volunteer’s lumbar spine with the back support was confirmed. A bench was provided to support the non-tested leg during rest periods to avoid excessive hip flexor stretching in supine position and discomfort while seated. A warming-up of submaximal isometric contractions was carried out for muscle and tendon pre-conditioning: 50%: 3 contractions; 75%: 2 contractions; and 90%: 1 contraction. Rest for 10 s was provided between submaximal contractions [8]. Following the warm-up, participants completed 12 MVICs and were encouraged verbally to cross their arms with hands on shoulders and to perform maximum strength on ramping contraction for 6–10 s and obtained visible feedback of the torque generated. A 2-min rest was provided between MVICs.

#### 2.5.2. EMG

The EMG of the VL, VM, RF, and most lateral portion biceps femoris was recorded bipolarly with a sampling frequency of 1000 Hz by the data accession device New Myotool (Miotec—Biomedical Equipment, Porto Alegre, Brazil^®^). The device was synchronized with the dynamometer, and electrical activity was recorded. Passive electrodes (circular silver-silver chloride electrodes with a 20 mm diameter) were positioned on the belly of the muscles [29] with an inter-electrode range (center to center) of 20 mm. A referred electrode was attached on the patella of the ipsilateral limb [30] Impedance reduction between the two electrodes was achieved through trichotomy and cleaning with alcohol. Raw EMG signal was band-pass filtered (20–500 Hz) to remove artifacts, and a notch filter of 60 Hz was applied. The raw RMS values were calculated within a 500 ms period in the most stable part of the torque trace (the MVIC plateau). Neuromuscular efficiency was calculated by dividing the peak knee extension torque by the RMS of the knee extensors [31] and then transformed into a percentage multiplied by 100.

#### 2.5.3. Muscle Architecture Assessment

We used an ultrasound system (M-turbo, Sonosite, Washington, USA) connected to a linear transducer (40 mm, 7.5 MHz, depth 6.0 cm, acquisition frame of 30 Hz). A water-based gel served as a coupling mean between the transducer and the skin surface. The muscle fibers were visualized at their longitudinal plane, being the transducer in a right angle with the skin at 50% (RF), 60% (VL), 75% (VM), and 80% (vastus intermedius [VI]), from cranial to distal, considering the thigh length (between the medial landmark of the anterior superior iliac spine and the patella base), as adjusted from prior comments [32,33]. These regions were chosen to allow a homogeneous muscle sonography, i.e., minimal fiber and constraints [32]. The RF and VI were scanned on the anterior thigh, while the VL and VM were scanned on the lateral and medial thigh surface, respectively. A customized Styrofoam apparatus retained the transducer avoiding undesired movement. The transducer alignment was also manually corrected to keep the superficial and deep aponeuroses in parallel, so several fascicles could be observed [34,35]. With the ultrasound set to record a 15-svideo, two recordings were made for each quadriceps femoris component during MVIC. The resting state was also recorded prior and after the MVIC trial. The recording with the best visualization of multiple fascicles was used for the measurement of Lf and θp. The video files were transferred to a computer for processing. Frames were selected at rest and at the MVIC plateau and stored as image files to be analyzed in ImageJ software (v. 1.46; National Institutes of Health, Bethesda, USA) (Figure 2). The fascicle that could be evidently delineated from the attachment point on the deep aponeurosis to the transducer field-of-view limits was used for the measurements [35]. The θp is the angle formed by the deep aponeurosis and the fascicle. The Lf is obtained by following the fascial from the superficial to the deep aponeurosis. When the fascicle was greater than the field-of-view boundaries, the insertion on the deep aponeurosis was kept, and the remaining portion up to the superficial attachment was estimated by [36]. For all ultrasound imaging results, the average of three measurements were used. A researcher with extensive experience in ultrasonography performed all measurements. In addition, we synchronized the MVIC and ultrasonographic recordings with a data acquisition device, New Miotool (Miotec Biomedical Equipment Ltd., POA, Brazil^®^; sampling rate: 2000 Hz, A/D converter: 14 bits, common rejection mode: 110 db at 60 Hz). The device was interfaced with the computerized dynamometer, and with a high-definition camera positioned to record the ultrasound system display. When the evaluator started the video ultrasound video recording, a visual indicator appeared on the ultrasound screen, which allowed the synchronization of all data on a torque-time trace generated on the New Miotool software [37].

#### 2.5.4. TAC Stiffness

The TAC displacement of RF, VL, VM, and VI were assessed using the same video recordings obtained for the muscle architecture variables. During data collection, a custom-made device held the probe, preventing it from moving. As mentioned above, care was taken to avoid the slide of the transducer on the skin surface. However, if sliding occurred, the TAC displacement was adjusted considering a hypoechoic shadow from an adhesive tape. Moreover, ultrasonographic recordings obtained (two for each muscle belly) during passive motion at 10°/s of the knee from 60° to 0° in both seated and supine positions were used to correct displacement overestimation due to any undesired angular rotation of the knee. Only the corrected values were used to calculate each constituent’s stiffness [22].

The Tracker 4.87 software allowed the manual tracking of the fascicle-deep aponeurosis attachment while it was displaced from rest to MVIC. If the deep insertion started outside the probe’s field-of-view, we made a linear extrapolation as previously described [21,25]. Quadriceps femoris muscle force was obtained by dividing the knee extensor torque by the patellar tendon moment arm, which was a fixed value according to the knee angle (60°: 0.056 m; 20°: 0.0475 m) [38]. To obtain a quadriceps femoris TAC stiffness of all quadriceps femoris constituents, we used the delta force from 50% to 100% of the MVIC divided by the mean delta displacement of each quadriceps femoris constituent also at 50% and 100% [33].

#### 2.5.5. Patellar Tendon Properties

For all analysis of patellar tendon properties, participants then performed six 5-s submaximal isometric knee extension MVICs for tendon pre-conditioning [39], which was mentioned above in the torque assessment section. Following the submaximal MVICs, the four MVIC were randomly performed to assess the patellar tendon with 120 s of rest between each. For patellar tendon variables, two volunteers were excluded due to technical problems in the analysis.

##### Morphological Properties

The same ultrasound system, settings, and synchronization method used for muscle architecture were also used for the assessment of the patellar tendon properties. The resting length was obtained with the ultrasound probe positioned longitudinally along the tendon, from the patella’s apex to the deep insertion to the tibial tuberosity [40]. If the size of the transducer did not allow the complete visualization of the patellar tendon, then it was obtained using an overlapping images method adopted by [41] (Figure 3). The length from the marker to each anatomical structure was measured with Tracker 4.87 software (www.physlets.org/tracker/ (accessed on 13 December 2018)) and summed to determine the patellar tendon length, according to [42].

Patellar tendon CSA was obtained with the ultrasound probe positioned perpendicular to the long-axis of the tendon. The mean value from three images was obtained at three locations (25%, 50%, and 75% of the tendon length) [43] to allow patellar tendon CSA to be measured from these axial images using Image J software (v. 1.46; National Institutes of Health, Bethesda, Maryland).

##### Mechanical Properties

Patellar tendon force was defined by the torque obtained during MVIC divided by the patellar tendon moment arm, determined from previous literature as 0.056 m and 0.0475 m at 60° and 20° of knee flexion, respectively [38]. Patellar tendon force was determined at 10% intervals of the MVIC (from 0 to 100%). The elongation was measured with cine-loop ultrasound imaging during MVIC using the same landmarks described above for the patellar tendon rest length. Patellar tendon elongation was defined as the length change between the patellar tendon proximal and distal insertions. The patella’s apex and the tendon’s deeper insertion to the tibial tuberosity were determined by manual tracking using Tracker 4.87 (www.physlets.org/tracker/ (accessed on 13 December 2018)). Patellar tendon strength and elongation were synchronized using the same technique mentioned above in the muscle architecture assessment section, as proposed by Bojsen-Moeller, 2003 [37]. Force-elongation plots were fitted with a second-order polynomial forced through zero. The slope of the stress-strain curve was used to calculate stiffness, based on the chosen force levels for measuring tendon stiffness. Linear regression was employed to derive the slopes of both the force-elongation and stress-strain curves, which were calculated from 50% to 100% MVIC [44].

##### Material Properties

Stresses and strains were obtained at 10% torque steps throughout the MVIC to assess the patellar tendon stress-strain relationship and estimate patellar tendon material properties for each condition [44]. The patellar tendon stress was calculated by dividing tendon force by CSA, and tendon strain was calculated by dividing tendon elongation by patellar tendon resting length. Stress-strain plots were fitted with a second-order polynomial forced through zero. Using the associated quadratic equations, Young’s modulus was determined as the stress-strain relationship using the same relative (50–100%) force levels as selected for determining tendon stiffness force.

### 2.6. Statistical Analysis and Sample Size

All outcomes are reported as geometric mean and 95% confidence intervals (95% CI). To compare peak knee extension torque, RMS, efficiency neuromuscular, tendon-aponeurosis complex stiffness, and patellar tendon properties between the different positions, we used repeated-measures one-way analysis of variance (ANOVA). For θp and Lf, once we had valued at rest and during MVIC, a repeated measure two-way ANOVA [position by condition (rest and MVIC)] was used. When a significant difference was detected, a Tukey post-hoc test was applied to identify the differences. Effect sizes and statistical power were calculated. The effect size was determined using partial eta squared (η*_ρ_*^2^), according to the following classification: small (η*_ρ_*^2^ = 0.01), medium (η*_ρ_*^2^ = 0.06), and large (η*_ρ_*^2^ = 0.14) effects [45]. 

For reliability assessment, the intra-class correlation (ICC) of torque (all eight MVIC performed during muscle ultrasound imaging for each position) was obtained using a mean of multiple measurements, absolute agreement, 2-way mixed-effects model. The purpose of this ICC was to guarantee that all of the muscle structure assessments were performed under stable conditions of contraction intensity. Moreover, a single-measurement, absolute-agreement, 2-way mixed-effects model was used for the interrater ICC of muscle architecture and the TAC displacement (two repeated analyses, seven to 14 between-days, of 25 recordings for each quadriceps femoris constituents). To determine the reliability of measuring tendon elongation, two repeated measurements of 25 random points (i.e., at any force level) were obtained for each condition from the force-elongation curve and used to calculate the ICC using a single-measurement, absolute-agreement, 2-way mixed-effects model. Reliability was classified as: poor (<0.5), moderate (0.5–0.75), good (>0.75–0.9), and excellent (>0.9). All statistical analyses used a significance level at *p* ≤ 0.05. All analyses were performed using STATISTICA 23.0 (STATSOFT Inc., Tulsa, Oklahoma, USA), and the software GraphPad PRISM 8.4.1 (San Diego, CA, USA) was used for graphic design.

The sample size was determined a priori using G*Power (version 3.1.3; University of Trier, Trier, Germany) with the level of significance set at *p* = 0.05 and power (1-β) = 0.80 to detect a large effect size (η*_ρ_*^2^ = 0.45). Based on Lanza et al. [8], we expected means and standard deviations from knee extension torques to be approximately 125.93 ± 31.81 Nm, 249.3 ± 30.13 Nm, 267.1 ± 32.26, and 216.3 ± 36.25 Nm for knee flexion angles of 25°, 50°, 80°, and 106°, respectively. Based on these values, we found a combined standard deviation of 63.62 Nm with a sample size of 20 participants.

## 3. Results

### 3.1. Reliability of Measurements

High test-retest reliability was observed from the ICC values for torque at SUP60 (0.92), SIT60 (0.94), SUP20 (0.92), and SIT20 (0.93). We obtained good reliability for the *θ*_p_ of RF (0.75), VL (0.78), VM (0.82), and VI (0.77), *L*_f_ of RF (0.81), VL (0.80), VM (0.77), VI (0.79), and tendon-aponeurosis complex displacement for RF (0.98), VL (0.95), VM (0.95), and VI (0.86) and for maximal elongation of the patellar tendon (0.98).

### 3.2. MVIC, Raw RMS, and Quadriceps Femoris Neuromuscular Efficiency

A significant main effect of position was found for peak knee extension torque (F 3, 57 = 87.57, *p* < 0.001, η*_ρ_*^2^ = 0.82, power = 1.0). The post-hoc analysis showed that knee flexed at 60° (SUP60 and SIT60) had higher MVIC (*p* < 0.001 for all analyses) than SUP20 and SIT20 (Figure 2). There was a non-significant main effect of position for raw RMS (F 3, 57 = 0.87, *p* = 0.460, η*_ρ_*^2^ = 0.04, power = 0.22) (Figure 4). A significant main effect of position was found for quadriceps femoris neuromuscular efficiency (F 3, 57 = 22.32, *p* < 0.001, η*_ρ_*^2^ = 0.54, power = 1.0). The post-hoc analysis showed that knee flexed at 60° (SUP60 and SIT60) had higher values (*p* < 0.001 for all analyses) than SUP20 and SIT20 (Figure 4).

For RF (Figure 5A,B), there was interaction between position and condition for the *θ*_p_ (F 3, 57 = 3.65, *p* = 0.017, η*_ρ_*^2^ = 0.16, power = 0.77). The post-hoc analysis showed that both at rest and during contraction, SUP60 had lower *θ*_p_ compared to SIT60, SUP20, and SIT20 (*p* < 0.001–0.036), with no differences between other comparisons (*p* = 0.15–0.99). There was no interaction of factors for *L*_f_ (F 3, 57 = 1.87, *p* = 0.140, η*_ρ_*^2^ = 0.089, power: 0.46), but the effect of position was significant (F 3, 57 = 24.89, *p* < 0.001, η*_ρ_*^2^ = 0.56, power = 1.00), where the post-hoc analysis showed greater Lf for SUP60 (*p* < 0.001; Figure 5B) than all positions, with no differences between other comparisons (*p* = 0.46–0.97).

The VL (Figure 5C,D) presented a significant interaction between positioning and condition for θp (F 3, 57 = 3.48, *p* = 0.021, η*_ρ_*^2^ = 0.15, power = 0.75). The post-hoc analysis showed that, at rest, there was lower θp for SUP60 compared to SUP20 (*p* = 0.012; Figure 5C) and SIT20 (*p* < 0.001), and at SIT60 compared to SIT20 (*p* = 0.033), with no differences between other comparisons (*p* = 0.31–0.97). Furthermore, during MVIC, *θ*_p_ was lower (*p* < 0.001) at SUP60 and SIT60 compared to SUP20 and SIT20. No significant differences were observed between SUP60 and SIT60 (*p* = 1.0), nor between SUP20 and SIT20 (*p* = 0.16). Position factor was significant for the *θ*_p_ (F 3, 57 = 13.66, *p* < 0.001, η*_ρ_*^2^ = 0.41, power = 0.99). The post-hoc analysis indicated lower θp (*p* < 0.001) for SUP60 and SIT60 compared to SUP20 and SIT20. No significant differences were observed between SUP60 and SIT60 (*p* = 0.90), nor between SUP20 and SIT20 (*p* = 0.37). There was no interaction for Lf (F 3, 57 = 0.56, *p* = 0.064, η*_ρ_*^2^ = 0.02, power = 0.15), but there was a significant main effect of positioning (F 3, 57 = 14.10, *p* < 0.001, η*_ρ_*^2^ = 0.42, power = 0.99), with post-hoc analyses showing higher *L*_f_ at SUP60 compared to SUP20 and SIT20 (*p* < 0.001), respectively. The same was true at SIT60 when compared to SUP20 and SIT20 (*p* < 0.001), respectively.

Only for VM (Figure 5E,F), there was no significant interaction between position and condition for θp (F 3, 57 = 0.31, *p* = 0.812, η*_ρ_*^2^ = 0.01, power: 0.10) and for Lf (F 3, 57 = 0.85, *p* = 0.46, η*_ρ_*^2^ = 0.043, power: 0.22). However, position factor was significant for both θp (F 3, 57 = 37.40, *p* < 0.001, η_ρ_^2^ = 0.66, power: 1.00) and Lf (F 3, 57 = 13.06, *p* < 0.001, η*_ρ_*^2^ = 0.40, power: 0.99), with post-hoc analysis indicated lower *θ*_p_ (*p* < 0.001) and greater Lf (*p* = 0.002–0.037) for SUP60 and SIT60 compared to SUP20 and SIT20.

Regarding VI (Figure 5G,H), there was a significant interaction effect between position and condition for both θp (F 3, 57 = 2.82, *p* = 0.046, η*_ρ_*^2^ = 0.12, power = 0.64) and Lf (F 3, 57 = 6.24, *p* < 0.001, η*_ρ_*^2^ = 0.24, power = 0.95). The post-hoc analysis showed that, at rest, there was a lower θp for SUP60 compared to SUP20 (*p* = 0.003) and SIT20 (*p* = 0.02). During MVIC a lower θp was found for SUP 60 and SIT60 when compared to SIT20 (*p* = 0.008; *p* = 0.027, respectively). Other pairwise comparisons at rest and during MVIC were not significant (*p* = 0.26–0.99). A greater Lf was found at rest only for SUP60 and SIT60 (*p* < 0.0001 for all analyses) when compared to SUP20 and SIT20. For Lf, other pairwise comparisons during rest were not significant (*p* = 0.56–0.99). However, during MVIC, Lf was greater only at SUP60 compared to SIT20 (*p* = 0.005). The main effect of position was also significant for both θp (F 3,57 = 4.40, *p* = 0.007, η*_ρ_*^2^ = 0.18, power = 0.85) and Lf (F 3, 57 = 16.16, *p* < 0.001, η*_ρ_*^2^ = 0.45, power = 0.99). The θp was lower at SUP60 compared to SIT20 (*p* < 0.005) and, Lf was greater for SUP60 and SIT60 compared to SUP20 and SIT20 (*p* < 0.001 for all analyses).

### 3.3. TAC Stiffness

The TAC stiffness of quadriceps femoris is presented in Table 1. A significant main effect of position was found for TAC (F _3,57_ = 7.84, *p* = 0.001, η*_ρ_*^2^ = 0.29, power = 0.98) and the post-hoc analysis showed that TAC stiffness was greater in SUP60 (*p* = 0.001), SIT60 (*p* = 0.0004) and SUP20 (*p* = 0.01) compared to SIT20.

### 3.4. Tendon Properties

The patellar tendon (morphological, mechanical, and material) properties for each position are presented in Table 1. The tendon force-elongation (A) and stress-strain relationships (B) are shown in Figure 6.

#### 3.4.1. Morphological Properties

No changes were found in the patellar tendon resting length (*p* = 0.186) and CSA (*p* = 0.563) for all conditions.

#### 3.4.2. Mechanical Properties

The mechanical properties of the patellar tendon are shown in Table 1 and Figure 4A. A significant main effect was found for patellar tendon force (F 3,51 = 33.90; *p* < 0.01; η*_ρ_*^2^ = 0.66; power = 1.00). In the post-hoc analysis both SUP60 and SIT60 showed a greater force (*p* < 0.001) than SUP20 and SIT20, with no differences between positions with the same knee angle: SUP60 vs. SIT60: *p* = 0.057; SUP20 vs. SIT20: *p* = 0.93. Maximal tendon elongation presented main effect (F 3,51 = 3.29; *p* = 0.027; η*_ρ_*^2^ = 0.16; power = 0.71) and the post-hoc analyses showed SIT20 significantly higher than SUP20 (*p* = 0.022), but there were no significant differences in comparison to SUP60 and SIT60 (*p* = 0.10, *p* = 0.19), respectively. Significantly greater stiffness at SIT60 was found compared to SUP20 and SIT20 (F 3,51 = 6.88; *p* < 0.01; η*_ρ_*^2^ = 0.28; power = 0.96) with post-hoc analysis *p* < 0.001.

#### 3.4.3. Material Properties

The material properties of the patellar tendon are shown in Table 1 and Figure 6B. The stress at SUP60 and SIT60 was significantly higher than at SUP20 and SIT20 (F 3,51 = 30.10; *p* < 0.01; η*_ρ_*^2^ = 0.63; power: 1.00). However, no differences were found in tendon stress at the same knee angle (SUP60 vs. SIT60 and SUP20 vs. SIT20). The tendon strain was not changed (*p* = 0.057). We found a significant main effect for Young’s modulus (F 3,51 = 7.01; *p* < 0.01; η*_ρ_*^2^ = 0.29; power: 0.97). In the post-hoc analysis, SIT60 was higher than SUP20 (*p* = 0.001) and SIT20 (*p* = 0.001).

## 4. Discussion

To the best of our knowledge, this is the first study to assess different hip and knee joint angles on torque generation, RMS activity, neuromuscular efficiency, muscle architecture, and tendon-aponeurosis complex stiffness of the quadriceps muscle constituents and patellar tendon properties in healthy adults. In general, we found: (1) higher torque and neuromuscular efficiency at 60° of knee flexion compared to 20°, regardless of hip position; (2) no differences for RMS between positions; (3) RF showed a lower pennation angle and greater fascicle length at SUP60 compared to all other positions, while VL, VM, and VI showed lower pennation angle_,_ and greater fascicle length at 60° of knee flexion when compared to 20°; (4) the TAC stiffness was greater at the more elongated position; and (5) tendon force, stiffness, stress and ‘Young’s modulus were greater with the knee flexed at 60°, compared to 20°.

### 4.1. MVIC, RMS, and Quadriceps Neuromuscular Efficiency

We found greater MVIC at 60° of knee flexion compared to 20°. According to Lanza et al. 2017, the differences in torque production are due to the force-length relationship of the muscle, in which changes in the joint’s angle and the muscle’s length affect the extent of force generation [8]. Thus, the knee extensor torque reduction on positions closer to the full extension could be partly attributed to mechanical factors, such as the reduced number of cross-bridges attached subsequently to sarcomere beyond the optimal actin-myosin overlap [6,14].

Our results demonstrated no differences in MVIC torque between supine and seated positions and corroborated Bampouras et al. (2017) [46]. In contrast, Maffiuletti and Lerpes, (2003) [11], and Ema et al. (2017) [12], found higher torque values in the seated position, which may be due to the difference in the operated region of the force-length relationship of RF between the two hip positions [12]. However, the choice of knee angle for Maffiuletti and Lerpes, (2003) [11], and Ema et al. (2017) were 90° and 70°, respectively. It is possible that we did not find any differences in our study since 60° of knee flexion may not have been enough to lengthen the RF and generate a considerable effect on torque output, showing a disadvantage from one position to the other [12].

We demonstrated no differences in RMS activity between positions. Babault et al. (2003) found higher activation values at short (i.e., 35° knee flexion) compared with long (i.e., 75° knee flexion) quadriceps muscle length [16]. With a shortened position, lesser muscle activation was expected [47,48]; higher activation was observed that would compensate for the weaker torque observed at higher degrees of knee flexion [16]. Maffiuletti and Lerpes, (2003) demonstrated greater activation in the seated position in comparison with the supine position for VM, VL, and RF muscles [11]. However, it is noteworthy that Maffiuletti and Lerpes, (2003), found the greatest neural activation of the knee extensors with the knee positioned at 90°, which may reflect a neurophysiological mechanism as compensation for the neuromuscular transmission-propagation deficiency and/or mechanical disadvantage of RF in a shortened position [11]. These results still fluctuate widely between these two positions because the lack of significant effect of the hip joint angle on agonist and antagonist muscle activations found by Ema et al. (2017) [12] suggests that neural factors may not have a substantial effect on the difference in knee extension torque and need to be further investigated.

Neuromuscular efficiency could be shown in several in vivo human studies, indicating optimized muscle function [49]. Aragão et al. (2015) consider those individuals as sufficiently capable of producing greater strength with a lower magnitude of muscle activation [50]. We found greater efficiency for the quadriceps femoris muscle in positions with the knee flexed at 60°. Although the RMS did not present differences between the positions, 60° positions indicate an economic and efficient mechanism since it was not necessary to increase muscle activation to generate greater torque, demonstrating the mechanical advantage of this joint angle.

### 4.2. Muscle Architecture

The observable adaptations in the muscular architecture during a contraction are the increase of the muscle thickness and the pennation angle and the decrease of the fascicle length, which are determinants in the generation of strength, range of motion and velocity of muscular shortening [17,51,52,53]. We found an increase in pennation angle of quadriceps femoris constituents from rest to contraction, consistent with previous studies [33]. Therefore, fascicle length and pennation angle change depending on the shortening or lengthening of the sarcomeres and the response to variations in tendon slack and total muscle length. As a result, these changes have important functional relevance concerning the production of force that is modified by the sarcomere and changes in whole muscle length [54].

We demonstrated an apparent effect of the hip angle on RF architecture, as expected for the quadriceps femoris’ biarticular constituent. The pennation angle was lower, and fascicle length was higher at SUP60 than in all other positions. The VL, VM, and VI operated with lower pennation angle when the knee was flexed at 60° compared to 20°. Placing the quadriceps femoris in a better physiological architectural configuration for generating torque favors a better transmission of muscle strength to the tendon and the ideal length of the sarcomere/fiber [55,56]. Furthermore, our findings demonstrate that fascicle length was shorter during VI contractions at SIT20, and the larger shortening would have been caused by taking up the elongated series elastic component [17]. Therefore, positions at 60° set the quadriceps femoris in a better architectural configuration, leading to a neuromuscular economy. This can be included in the proposals for strength rehabilitation programs, since an improvement was observed in the neuromuscular transmission of muscle strength to the tendon.

### 4.3. TAC Stiffness

The TAC stiffness index of quadriceps femoris on SUP60 was higher than on all other positions, similar to other studies [42], indicating an increased passive tension that limited tendinous elongation during contraction [24]. Shortened positions limit the mechanical stress and consequently lead the muscle to bear less force and generate less stress on the tendon. The increased tension of the TAC in stretched conditions is known to allow stronger contractions with less effort due to better force transmission [26,27].

### 4.4. Patellar Tendon Properties

#### 4.4.1. Morphological Properties

Patellar tendon resting length and CSA did not differ between conditions. Similar results were previously observed considering the changes in knee angle [57,58]. We showed that the hip angle, from 85° of flexion to 0°, also did not provide any lengthening of the patellar tendon. This probably occurred because tendons designed to withstand high forces should not suffer significant length change between relatively close knee angles (60° and 20°), even with the additional stretch promoted by the hip extension [42]. The lack of changes in patellar tendon length may reflect biomechanical implications since it is not likely to attribute differences in stiffness to appreciable changes in the resting length, but possibly to collagen molecule coiling/uncoiling [59], associated with crimp pattern, which may imply transmission of force and load in tendons [60].

#### 4.4.2. Mechanical and Material Properties

Stiffness presented higher values at SIT60 in comparison with SUP20 and SIT20 positions. It is probably due to higher levels of force being applied to the tendon and, consequently, the higher level of tendon stiffness presented. It is possible to notice that a longer position generates greater tendon stiffness, in agreement with Kubo et al. (2006) [22] during an isometric training protocol. According to these results, it seems preferable to load the patellar tendon at voluntary contractions using the knee at 60°. Simultaneously, the hip angle variation may affect how tensile loads are transmitted through the tendon. The increase of stiffness in positions at 60° can provide an advantage in rehabilitation since it promotes more significant tension generation in the muscle-tendon unit. As tendon stiffness increases with high-intensity training [56,61], the high load achieved by muscle contractions in this angle can lead to higher tendon adaptations than training programs with lower loads. A remarkable finding was that patellar tendon stress supports the force results (i.e., significant stress with the knee at 60° without the hip’s influence). Therefore, we cannot attribute these results to differences in the CSA average, but rather to the different higher strength levels in positions at 60° of knee flexion. Stress and Young’s modulus were greater with the knee flexed at 60° compared to 20°.

A potential limitation of this study was the lack of estimated contributions of each muscle for the total quadriceps muscle force. Therefore, tendon-aponeurosis complex stiffness of each quadriceps muscle constituent was calculated considering the total force. These calculations may lead to errors due to changes in contribution according to both force and muscle length levels. However, if we do not perform comparisons between the constituents, our values may be useful as a snapshot. Another limitation was restricted to the healthy young male population used in this study. More broadly, research is also needed to determine these properties in clinical populations and other muscles and tendons. Finally, although the participants were carefully examined (verbal interview, visual inspection, palpation, passive and active movement) before inclusion in the study to check for any visible musculoskeletal abnormality, we did not perform objective measures that could better inform the volunteers’ physical characteristics, such as the Q angle, somatotype, and arches of the foot. This information may be important for future research to better qualify the healthy populations and improve understanding of their specificities.

## 5. Conclusions

Torque generation, neuromuscular efficiency, a greater fascicle length and lower pennation angle, the patellar tendon force, stiffness, stress and ‘Young’s Modulus were higher with the knee flexed at 60° compared to 20°. Elongation was higher at SIT20 compared to the SUP20 position. All quadriceps femoris constituents presented higher tendon-aponeurosis complex stiffness in more elongated positions, indicating a higher capacity to support tension and expose the tendon to greater stress. In this way, our results suggest the superiority of the knee angle at 60° for isometric contractions compared to 20° comes with significant physiological and structural characteristics, which may be important factors guiding the adaptation to regular training/rehabilitation on the muscle-tendon unit. Furthermore, the hip angle was involved in changes in the quadriceps muscle (not only the rectus femoris) which may be explored in further studies. These findings are essential for understanding the quadriceps femoris muscle-tendon unit’s behavior in detrimental hip-knee angle positions and should be brought to the attention of rehabilitation programs since they could be related to force transmission. It is possible to suggest that clinicians preferably use SUP60 or SIT60 conditions for strengthening and remodeling purposes since these positions seemed to provide a mechanical advantage for generating greater strength. Gaining a better understanding of the possible physiological mechanisms that underlie muscle and tendon efficiency can provide a framework to develop strengthened protocols to produce more effective contractions and improve the outcomes of rehabilitation programs.

## Figures and Tables

**Figure 1 ijerph-20-03947-f001:**
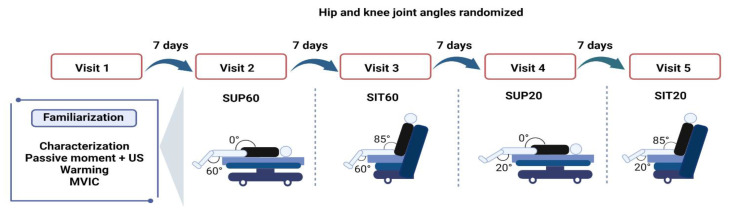
Design of the experimental conditions: participants engaged in five designed sessions at least seven days apart, a familiarization and four experimental sessions to randomly test four compositions of hip and knee joint angles during quadriceps femoris musculature (QF). Twelve MVICs were required at each visit. Legend: MVIC: Maximal Voluntary Isometric Contraction; SUP60: supine with 60° of knee flexion; SIT60: seated with 60° of knee flexion; SUP20: supine with 20° of knee flexion; SIT20: seated with 20° of knee flexion.

**Figure 2 ijerph-20-03947-f002:**
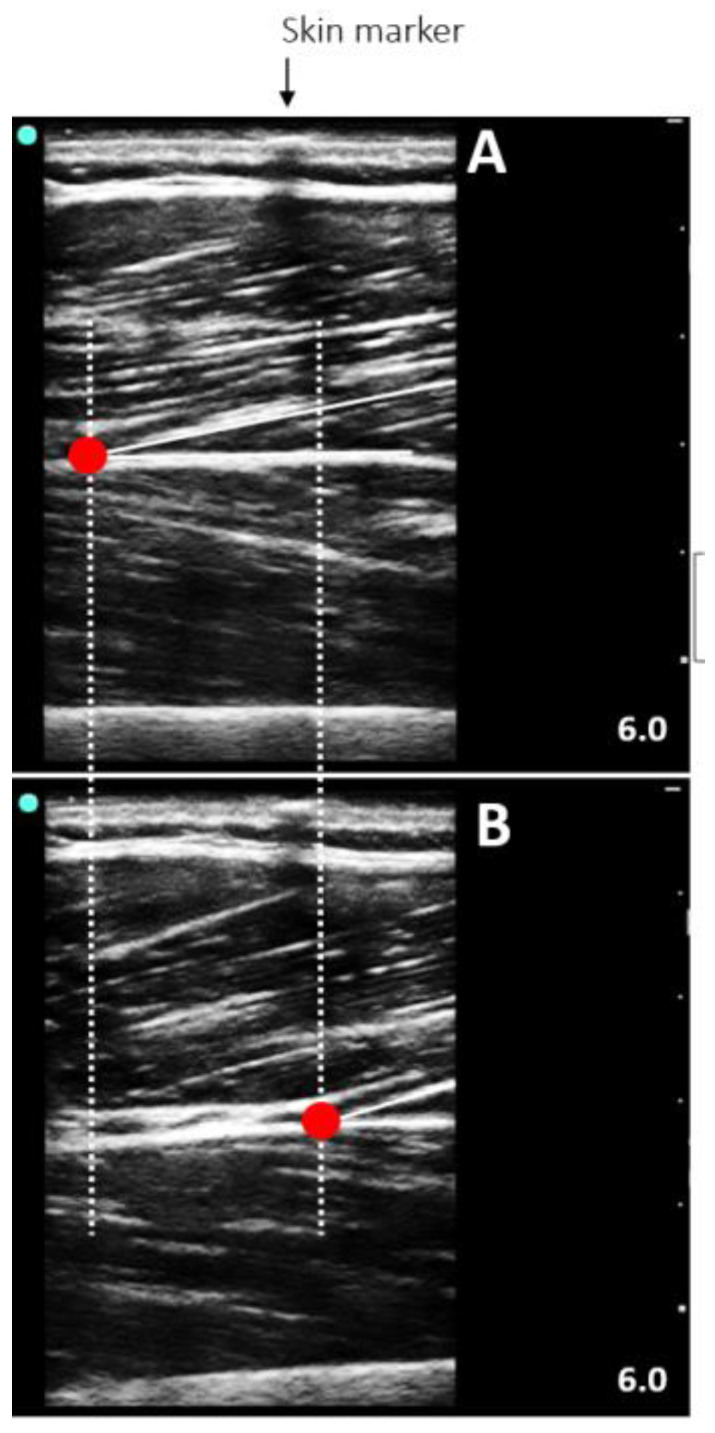
Representative ultrasound image of the vastus lateralis at rest (**A**) and during a maximum voluntary isometric contraction (**B**) in a seated position with the knee flexed at 60°.

**Figure 3 ijerph-20-03947-f003:**
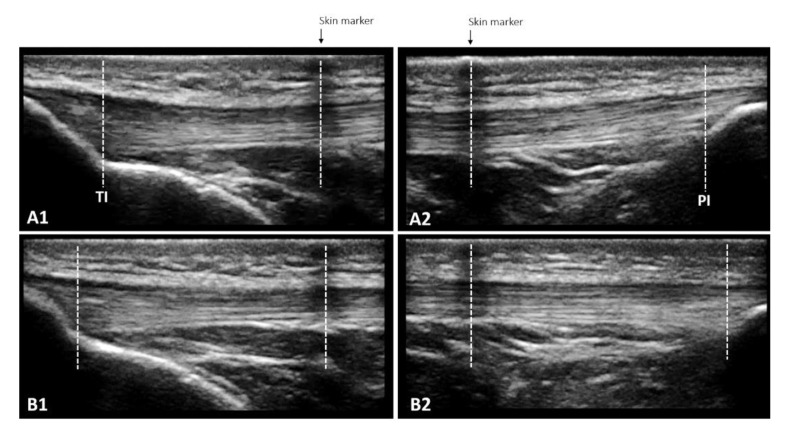
The overlapping images technique enables the measurement of patellar tendon length at rest (**A1** + **A2**) and in increments of 10% force up to maximum voluntary isometric contraction (**B1** + **B2**) when the entire length cannot be captured in a single frame due to the limited size of the ultrasound probe. A skin marker (adhesive tape) that creates a hypoechoic shadow is used to define the measurement bounds: the length from TI to the center of the skin marker in **A1** and the length from the center of the skin marker to PI in **A2** are added. The same procedure is repeated for each 10% increase in force, leading to **B1** and **B2**.

**Figure 4 ijerph-20-03947-f004:**
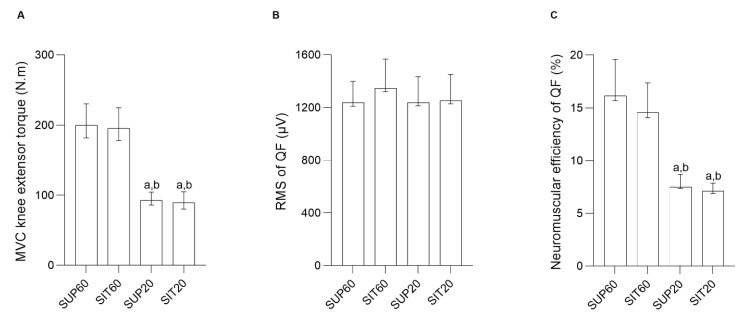
MVIC knee extensor torque, Root Mean Square and neuromuscular efficiency of the quadriceps femoris constituents according to hip and knee angles during contraction. (**A**) MVIC knee extensor torque (N.m) (left *y*-axis) and different conditions (right *x*-axis). (**B**) Root Mean Square of quadriceps femoris (µV) (left *y*-axis) and different conditions (right *x*-axis). (**C**) Neuromuscular efficiency of the quadriceps femoris (%) (left *y*-axis) and different conditions (right *x*-axis). Data are presented as geometric means and confidence intervals (CI 95%). Legend: *MVIC:* Maximal Voluntary Isometric Contraction; *RMS:* Root Mean Square; *QF:* Quadriceps femoris; *SUP60:* supine with 60° of knee flexion; *SIT60:* seated with 60° of knee flexion; *SUP20:* supine with 20° of knee flexion; *SIT20:* seated with 20° of knee flexion. Significant differences: ^a^ different from SUP60 at (*p* ≤ 0.05). ^b^ different from SIT60 at (*p* ≤ 0.05).

**Figure 5 ijerph-20-03947-f005:**
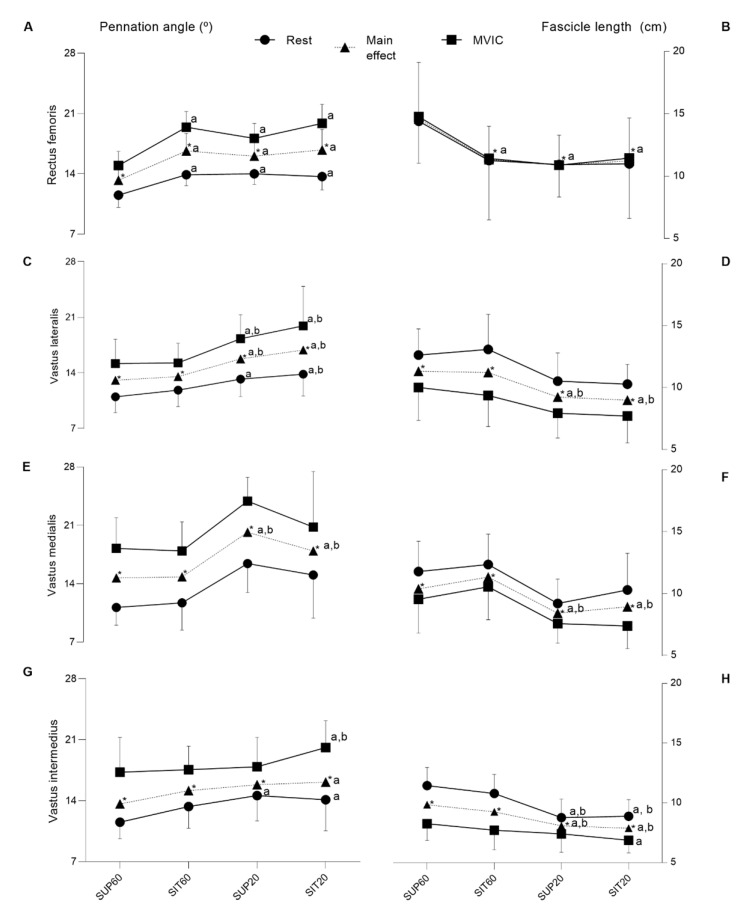
The pennation angle and fascicle length of the quadriceps femoris constituents according to hip and knee angles during rest and contraction. Muscle (left *y*-axis) and different conditions (right *x*-axis). Rest (closed circle line), main effect (closed triangle line) and during MVIC (closed square line). The first column (pennation angle—θp (°) and second column (fascicle length—Lf (cm)). Data are presented as means and confidence intervals (CI 95%). (**A**,**B**) Rectus femoris; (**C**,**D**) Vastus lateralis; (**E**,**F**) Vastus medialis; (**G**,**H**) Vastus intermedius. Legend: *SUP60:* supine with 60° of knee flexion; *SIT60:* seated with 60° of knee flexion; *SUP20:* supine with 20° of knee flexion; *SIT20:* seated with 20° of knee flexion; θp: pennation angle; Lf: fascicle length. Significant differences: ^a^ different from SUP60 at (*p* ≤ 0.05); ^b^ different from SIT60 (*p* ≤ 0.05); * indicate significant differences in intensity (*p* ≤ 0.05) between rest and MVIC.

**Figure 6 ijerph-20-03947-f006:**
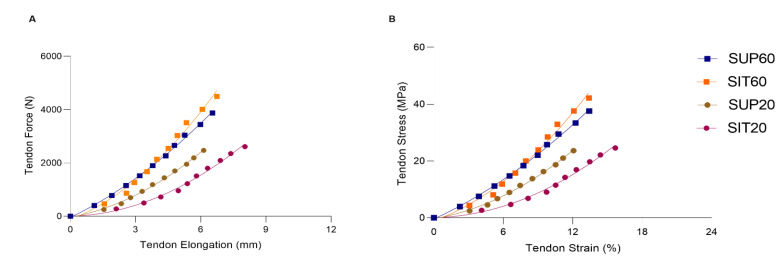
The patellar tendon force-elongation (**A**) and stress-strain (**B**) relationships according to hip and knee angles during MVIC. SUP60 (closed square blue line); SIT60 (closed square green line); SUP20 (closed circle red line) and SIT20 (closed circle purple line). Data are presented as mean (SD). Legend: *SUP60:* supine with 60° of knee flexion; *SIT60:* seated with 60° of knee flexion; *SUP20:* supine with 20° of knee flexion; *SIT20:* seated with 20° of knee flexion.

**Table 1 ijerph-20-03947-t001:** Patellar tendon properties and tendon aponeurosis complex stiffness of quadriceps femoris at different hip and knee angles measured during maximal voluntary isometric contraction. Geometric mean (95% CI).

	SUP60	SIT60	SUP20	SIT20
Morphological properties				
Resting length (mm)	48.95 (46.79–51.53)	50.45 (47.92–53.64)	50,62 (48.40–53.29)	50,55 (47.94–53.86)
CSA (mm^2^)	103.33 (97.63–110.45)	107.18 (102.23–113.16)	104.03 (98.20–111.14)	107.12 (103.98–110.68)
Mechanical properties				
Force (N)	3768.9 (3391.6–4353.1)	4341.37 (3913.85–5070.55)	2379.82 (2134.06–2807.60) ^a,b^	2497.16 (2193.69–3025.21) ^a,b^
Elongation (mm)	6.19 (5.44–7.64)	6.25 (5.49–7.98)	5.72 (5.07–7.20)	7.51 (6.53–9.54) ^c^
Stiffness (N/mm)	771.75 (636.78–1112.95)	1008.33 (822.89–1674.57)	600.05 (511.31–816.29) ^b^	579.92 (492.24–790.34) ^b^
TAC Stiffness QF (N/mm)	174.17 (169.18–189.15)	176.22 (171.19–195.83)	158.77 (154.35–179.66)	114.00 (111.28–130.51) ^a,b,c^
Material properties				
Stress (MPa)	36.48 (32.61–42.42)	40.51 (36.32–47.92)	22.88 (20.65–26.62) ^a,b^	23.43 (20.61–28.53) ^a,b^
Strain (%)	12.68 (11.15–15.69)	12.40 (10.83–15.97)	11.29 (9.97–14.14)	14.85 (13.02–18.34)
Young’s Modulus (Mpa)	393.65 (320.31–589.37)	512.28 (434.07–845.60)	313.63 (271.71–426.35) ^b^	303.84 (256.47–416.28) ^b^

Legend: CI: confidence interval; SUP60: supine with 60° of knee flexion; SIT60: seated with 60° of knee flexion; SUP20: supine with 20° knee flexion; SIT20: seated with 20° of knee flexion; CSA: Cross-sectional area, Stiffness slope of the force-elongation curve from 50 to 100% of maximal voluntary contraction force; TAC: Tendon aponeurosis complex; QF: Quadriceps femoris; Young’s modulus slope of the stress-strain curve obtained from 50 to 100% of maximal voluntary stress. ^a^ Significantly different from SUP60 at *p* < 0.05. ^b^ Significantly different from SIT60 at *p* < 0.05. ^c^ Significantly different from SUP20 at *p* < 0.05.

## Data Availability

The data presented in this study are available on request from the corresponding author. The data are not publicly available due to restrictions of a private or ethical nature and reasons of sensitivity, e.g., human data, participants’ location.

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
