# Peer review of "The Influence of Hip and Knee Joint Angles on Quadriceps Muscle-Tendon Unit Properties during Maximal Voluntary Isometric Contraction"

_ijerph, 2023, doi:10.3390/ijerph20053947_

Round 1

Reviewer 1 Report (Previous Reviewer 1)

Dear Authors. Thank you for considering my comments and those of my fellow reviewers. After corrections and addition of suggested information and photos, the article became clear and complete. I recommend the article for publication.

Best regards.

This manuscript is a resubmission of an earlier submission. The following is a list of the peer review reports and author responses from that submission.

Round 1

Reviewer 1 Report

The study is well performed and interesting to read. However, I have a few reservations:
1. I
nstead of performing ANOVA, I would expect to use mixed-effects models (LMM).
2. T
here is no justification for using a parametric test
3. W
hen reporting p-values ​​in post-hoc tests, the values ​​adjusted for multiple comparisons should be reported (padj)
4.
p≤0.050 is written in several places (should be p<0.050)
5.
correct the notation of the effect size ηρ2 (it is better to use subscripts and the correct letters: ηp2
6.
it would be good to indicate the method to calculate the corrected value padj.
7.
reporting of ANOVA results should be a bit standardized (for this structure: F (3, 57) = 0.87, p = 0.460, ηρ2=0.04, power = 0.22)
8.
note should be corrected (for this: level of significance set at α =0.05 )
9.
In the article, I would like to find representative ultrasonography photos, because you are writing about the use of US to study the assessment of muscle contraction. Such photos would be very valuable.

Author Response

Dear reviewer, first and foremost, thank you for your valuable comments. We have tried to address them to the best of our capabilities, and we believe the manuscript has significantly improved based on your suggestions. Your questions were addressed point-by-point and are provided below.

Sincerely yours,

Rita de Cassia Marqueti  Durigan

Associate Professor

Director of Muscle and Tendon Plasticity Research Group

PI - Laboratory of Molecular Analyses

Universidade de Brasília

Reviewer 2 Report

Thank you very much for the opportunity to review this article. The hypothesis is correct and the topic is very important. I will point out a few elements that, in my opinion, should be supplemented or improved.

1. I don't like the sentences in lines 105-106 that indicate the clinical goal for patients with knee joint injuries. Healthy people are indicated in the main inclusion criterion. Maybe it is worth taking care of healthy people at this stage of work, and then, in relation to this work, to examine people with knee injuries.

2. From the point of view of examining the lower limbs, the description of the material is insufficient We know a lot about the behavior before the tests. This information appears in various places. However, we do not know whether there was valgus of the knees, what was the Q angle (the angle between the thigh and the lower leg). We do not know the state of musculature or fatness of the limbs. We don't know the somatotype. This is important from the point of view of force vectors, muscle shaping, and tissue visibility. We know nothing about the arches of the foot. This may be less important, but for this type of research project, it is worth describing it.

3. I didn't notice the description about body stabilization during the tests. Especially, how was lordosis control taken care of? This is important because it can cause pain. Was it explained how to stabilize lordosis? There may be a paradox in which the lumbar section is overloaded during the assessment of the knee joint.

4. Did the subjects have upper limbs? If so, and I assume so, could they hold on to the chair, were there any special grips, did they keep their hands, for example, on the chest. I'm sorry, but the figures do not show the upper limbs, hence my question.

5. Whether the seat of the chair had adjustable length in the seat. In the supine position, this is not necessary. People with an average body height of 177 ± 6.3 cm took part in the study. Thus, the study included people whose body height ranged between 170 and 183 cm. Explain whether the standardized thigh seat length could have influenced the results of the study.

6. In chapter 2.4 I did not find information about the limb that was examined. Left or right? But also dominant or non-dominant? I think it is worth doing this type of analysis, especially since it may turn out that in both cases you will get different information. I don't know, but with such a high placement of the article, it is important to at least mention that the subject has two lower limbs, different not only because of the side of the body.

7. In my opinion, the content from chapter 3.1 should be included in the statistics chapter.

I guess that's it. The rest seems to be properly prepared.

Author Response

Dear reviewer, we would like to thank you for the comments and the manuscript's careful revision. We have tried to address all of them and believe that the manuscript has improved significantly based on your suggestions. 

Sincerely yours,

Rita de Cassia Marqueti  Durigan

Associate Professor

Director of Muscle and Tendon Plasticity Research Group

PI - Laboratory of Molecular Analyses

Universidade de Brasília

Reviewer 3 Report

Thank you for the opportunity to review this paper. I think the authors performed an interesting and important study. However, the paper needs work before I would recommend it for publication. My first recommendation is that the paper is divided into 2 separate papers. I suggest writing separate papers, one presenting the muscle activation and the other the mechanical properties of the muscle and tendon. I think that presenting the information together is confusing and detracts from the results of the study. I have made a few specific comments below.

The introduction needs to be rewritten for clarity and to improve the strength of the author’s rationale for their work. The authors present a strong hypothesis for the work, but the introduction is difficult to follow. There are several grammatical issues throughout the paper that need to be corrected.

Line 42 “as we as” correct

Line 43 “whichemains” correct

Lines 60-61 this sentence does not advance the author’s argument

Lines 61-66 These sentences read more like an introduction to the authors’ second paragraph

Lines 95 “we aimed to influence of the hip” This sentence does not make sense, suggest rewriting.

Methods lines 187 – 223 I recommend the authors provide photographs and sample US images to help readers know how the US images were collected and analyzed.

Lines 224 – 243  I suggest providing more information about how the tendon stiffness was calculated. I again suggest that photos and sample US images will help readers understand how the stiffness was measured.

Lines 251- 264 The authors use different citation styles.

Lines 289-299 I am not sure about the statistical analysis the authors performed. I looks to me like they used a mixed model three-way repeated measures ANOVA (hip position x knee position x contraction)

Author Response

We thank the revisor for kindly pointing out the several corrections needed on the introduction. Based on your suggestions, we think we have significantly improved the writing quality, as well as the rationale. The new information are red-marked. To avoid a too-marked text, sentences that were just moved up or down on the paragraphs were not marked, just the new text. We only red-marked and traced (example) when significant texts were deleted. 

Sincerely yours,

Rita de Cassia Marqueti  Durigan

Associate Professor

Director of Muscle and Tendon Plasticity Research Group

PI - Laboratory of Molecular Analyses

Universidade de Brasília
